# A Novel Method for Identifying Essential Genes by Fusing Dynamic Protein–Protein Interactive Networks

**DOI:** 10.3390/genes10010031

**Published:** 2019-01-08

**Authors:** Fengyu Zhang, Wei Peng, Yunfei Yang, Wei Dai, Junrong Song

**Affiliations:** 1Faculty of Information Engineering and Automation, Kunming University of Science and Technology, Kunming 650093, China; zfy_10086@163.com (F.Z.); yangyf@escience.cn (Y.Y.); dw@cnlab.net (W.D.); 2Computer Center of Kunming University of Science and Technology, Kunming 650093, China; 3Faculty of Management and Economics, Kunming University of Science and Technology, Kunming 650093, China; sjunrong@gmail.com

**Keywords:** essential genes, dynamic network, network fusion, protein–protein interactive network

## Abstract

Essential genes play an indispensable role in supporting the life of an organism. Identification of essential genes helps us to understand the underlying mechanism of cell life. The essential genes of bacteria are potential drug targets of some diseases genes. Recently, several computational methods have been proposed to detect essential genes based on the static protein–protein interactive (PPI) networks. However, these methods have ignored the fact that essential genes play essential roles under certain conditions. In this work, a novel method was proposed for the identification of essential proteins by fusing the dynamic PPI networks of different time points (called by FDP). Firstly, the active PPI networks of each time point were constructed and then they were fused into a final network according to the networks’ similarities. Finally, a novel centrality method was designed to assign each gene in the final network a ranking score, whilst considering its orthologous property and its global and local topological properties in the network. This model was applied on two different yeast data sets. The results showed that the FDP achieved a better performance in essential gene prediction as compared to other existing methods that are based on the static PPI network or that are based on dynamic networks.

## 1. Introduction

Essential genes (and their encoded proteins) play an indispensable role in supporting the life of organisms, and without them, lethality or infertility is caused. Studying essential genes helps us to understand the basic requirements for cell viability and fertility [1]. Moreover, identifying the essential genes of bacteria contributes to finding potential drug targets for new antibiotics [2]. Recently, some researchers pointed out that essential genes have a close relationship with human diseases [3]. Studying essential genes also helps us to design novel strategies for disease therapy. However, the methods to experimentally discover essential genes in biology are time consuming and inefficient. Consequently, several recent computational methods have been proposed to identify essential genes [4,5]. Generally, these computational methods can be classified into three categories: sequence-based methods, network-based methods, and multi-biological information-based methods.

Sequence-based methods are based on the fact that essential genes evolve much slower than other genes, and that they usually conserve across different species [6,7,8]. This kind of method usually infers essential genes by comparing their sequences with the sequences of known essential genes in the same species or other species. However, with the exponential increase of sequencing gene data, few of them can find orthologous essential genes in the same species or in the species at a long evolution distance.

Network-based methods identify essential genes according to their topological properties in the protein–protein interactive (PPI) networks. Essential genes are supposed to be the center of the PPI network, because removing them from the networks would cause the lethality and break down of the network [9]. A series of genes’ centrality scores in the networks have been employed to identify the essential genes, including Degree Centrality (DC) [10], Betweenness Centrality (BC) [11], Closeness Centrality (CC) [12], Subgraph Centrality (SC) [13], Eigenvector Centrality (EC) [14], Information Centrality (IC) [15], and Edge Clustering Coefficient Centrality (NC) [16].

However, there are some limitations in the network-based methods, including incomplete and error-prone currently available PPI data, and the neglect of other intrinsic properties of the essential genes. To overcome these limitations, some methods integrate PPI networks with other biological information to improve the prediction accuracy of essential proteins. One of these methods refines the currently available PPI network by introducing other biological information, such as gene functional annotation data [17], gene expression data, a centrality measure by integrating protein-protein interaction data and gene expression data (PEC) [18], weighted degree centrality (WDC) [19]), both gene functional data and gene expression data [20], and subcellular information [21]. Another type of method predicts essential genes by comprehensively utilizing the inner biological properties and the topological properties of essential genes. For example, considering the conservative property of essential genes, Peng et al. [22] proposed a method for predicting essential genes by integrating the orthology with the PPI networks, namely ION. Considering the close relationship between genes’ essentiality and the protein domain of their coded proteins, the same research group [23] introduced protein domain information into the PPI network to predict essential genes. Since essential genes are highly likely to be rich in protein complexes, some methods use information on known protein complexes [24,25] to identify essential genes. Recently, researchers tried to improve the prediction performance of essential genes by combining more biological properties. Li et al. [20] combined subcellular localization information, orthology information, and PPI networks to predict essential genes. Zhang et al. [26] detected essential genes based on the network topology, gene expression data, and gene ontology information.

However, the aforementioned methods all ignore the fact that essential genes interact with each other and play essential roles under certain conditions. Most of these methods are based on the static PPI networks (called the S_PPI), which consist of interactions accumulated in different conditions and time points. In fact, the interactions between genes in cells change over time, environment, and different stages of the cell cycle [27]. Some researchers try to construct dynamic PPI networks by combining the S_PPI with gene expression profiles [28]. These methods firstly identify the active genes at each time point, according to their expression levels. After that, these active genes are mapped to the S_PPI and a serial of dynamic PPI networks (called the D_PPI) are constructed for each time point. Different methods use different strategies to select active genes at each time point. Tang et al. [29] select the active genes of each expression time point if their corresponding expression values are above a threshold. However, this method will filter out the active genes with low expression values. To overcome this shortcoming, Wang et al. [30] proposed a three-sigma principle-based method that sets a threshold for each gene by considering its own expression curve over different time points. Xiao et al. [4] divide the active genes into two categories: Time-dependent genes and time-independent genes. The active genes of each time point are identified using a k-sigma method, where the k-sigma method computes an active threshold for each gene according to the mean and standard deviation of its expression values. Li et al. [31] use both the subcellular information and the temporal information to construct a dynamical PPI network. Recently, some methods that detect essential genes based on the dynamical PPI networks have started to emerge. Xiao et al. [32] map all the active genes of each time point detected by their method in Reference [4] to a static PPI, and then predict the essential genes using some centrality methods, i.e., DC [10], NC [16], BC [11], CC [12], and SC [13]. Li et al. [33] refine the S_PPI according to whether or not the interactive genes are simultaneously active at a time point and locate in the same subcellular location. After that, more centrality methods besides the methods mentioned in Reference [32] are implemented on the refined networks to predict the essential genes. For all we know, these methods improve the prediction of essential proteins either by using active genes in the dynamic PPI network to refine the static PPI network [32,33,34], or by simply averaging the scores of the active genes at each time point to get final ranking scores. More complex strategies should be designed to predict the essential genes from dynamic PPI networks. 

In this work, a novel method was developed for the identification of essential genes by fusing the dynamic PPI networks of different time points (FDP). Firstly, a serial of active PPI networks of each time point was constructed using Xiao’s method [4], and then these active PPI networks were fused into a final network using the method similar to that described in Reference [35]. In contrast to the method in Reference [35], FDP fuses the active PPI networks one by one according to their similarities. The nodes in the final network are active for at least one time point and their interactions are similar across all the time points. This idea relates to previous observations that the mRNA expression levels of essential genes tend to be high (active) [36] and vary, on average, within a narrow range, whereas the expression of non-essential genes fluctuates more widely [37]. Finally, a novel centrality method was designed to assign a ranking score to each node in the final network, whilst considering its orthologous property and both its global and local topological properties in the network. FDP, as well as eleven other existing methods were applied to predict yeast essential genes. Prediction results showed that FDP not only outperformed the existing methods that were based on the static PPI network, but it also outperformed the methods that were based on the dynamic PPI network. 

## 2. Materials and Methods

### 2.1. Materials

FDP and other existing computational methods were applied to predict the essential genes of S. cerevisiae (Bakers’ Yeast). Two different PPI datasets of *Saccharomyceas cerevisiae* were adopted to evaluate our method. One dataset was the DIP_PPI, downloaded from the DIP database [38] published on 10 October 2010. There were a total of 5093 proteins and 24,743 interactions, excluding self-interactions and repeated interactions. The other dataset was the SC_net from Reference [39], which consisted of 4746 proteins and 15,166 distinct interactions. 

The list of essential genes was integrated from the following databases: The Munich Information Center for Protein Sequences (MIPS) [40], *Saccharomyces* genome database (SGD) [41], Database of Essential Genes (DEG) [42], and *Saccharomyces* Genome Deletion Project (SGDP) [43]. There were 1285 essential genes, where only 1167 essential genes present in the DIP_PPI network and 1130 essential genes present in the SC_net network. The yeast’s gene expression data came from Reference [28], including 6777 gene products under 36 different time points of three life cycles. Therefore, there were 2759 genes or 2559 genes that appeared in the dynamic PPI networks constructed by combining the yeast’s gene expression data with the DIP_PPI network or the SC_net network, respectively. Moreover, 827 of the 2759 genes in the dynamic DIP_PPI network were essential genes and 785 of the 2559 genes in the dynamic SC_net network were essential genes. Table 1 lists the detailed information of the two yeast data sets.

Information on the orthologous proteins was taken from Version 7 of the InParanoid database. In our study, yeast proteins were mapped to another 99 species to find their orthologous proteins. Only the proteins in the seed orthologous sequence pairs of each cluster generated by InParaniod were chosen as the orthologous proteins.

### 2.2. Methods 

Figure 1 illustrates the workflow of the FDP. FDP takes three main steps to predict essential genes. Firstly, active PPI networks of each time point were constructed using Xiao’s method [4]. After that, a fusion method similar to Reference [35] was adopted to fuse the active PPI networks of different time points, and then a final network was constructed, in which the nodes were active for at least one time point and their interactions were similar across all the time points. Finally, a novel centrality method was designed to assign a ranking score for each node in the final network, and the nodes ranked on top were selected as the candidate essential genes.

#### 2.2.1. Constructing Dynamic Protein–Protein Interactive Networks

Our dynamic PPI networks were constructed based on the gene expression profiles and the PPI network. The expression profiles consisted of periodically (time-dependent) and non-periodically (time-independent) expressed profiles and some inevitable noise. De Lichtenberg et al. [44] point out that periodically expressed genes are more likely to be dynamically deterministic than random. However, the non-periodically expressed genes are more likely to be random than dynamically deterministic. Therefore, the first step to construct the dynamic PPI networks was to detect the time-dependent genes and time-independent genes from the time-course gene expression profiles using an AR (autoregressive) model as in Reference [45]. 

Let *x* = {*x*_1_, …, *x_m_*, …, *x_M_*} be a time series of observation values at equally-spaced time points from a dynamic system. A gene is supposed to be time dependent if its gene expressions have linear relationships and can be modeled by an AR model of order p (see Equation (1)). A gene is regarded to be time independent if its gene expressions have nonlinear relationships and can be modeled by an AR model of order zero (see Equation (2)).
(1)$$x_{m}=\beta_{0}+\beta_{1}x_{m-1}+\beta_{2}x_{m-2}+\ldots +\beta_{p}x_{m-p}+\varepsilon_{m};m=p+1,\ldots ,M,$$
(2)$$x_{m}=\beta_{0}+\varepsilon_{m}$$
where *β_i_* (*i* = 0, 1, …, *p*) is the autoregressive coefficient, and *ε_m_* (*m* = *p* + 1,…, *M*) denotes the random error, which follows a normal distribution with a mean of 0 and a variance of σ^2^. Since the order of the AR model in Equation (1) is unknown, similar to Reference [4], the *p*-values for all possible orders *p* (1 ≤ *p* ≤ (*M* − 1)/2) were calculated. A gene is regarded to be time dependent if one of these *p*-values calculated from its expression profile is smaller than a user-preset threshold value (threshold = 0.01). The expression profiles of a gene will be considered as noise if the gene is not only time-independent, but also if the mean of its expression values across all time points is very small (less than 0.5, according to the analysis in Reference [4]).

After identifying the time-dependent and the time-independent genes, and filtering out the noisy genes, the next step was to detect which of them were active at each time point. A gene was considered to be active when its expression value was above a given threshold (see Equation (3)). In this work, similar to Reference [34], we set the threshold for each gene using the following k-sigma principle, where *k* was set to 2.5, *u* and *σ* were the mean and standard deviation of their expression values.
*Active threshold* = *u* + *kσ* × (1 − *F*)
(3)
(4)$$F\left(p\right)=\frac{1}{1+\sigma^{2}\left(p\right)}$$

Therefore, a serial of active PPI networks was generated by mapping the active genes at each time point to the S_PPI and extracting the edges connecting them. Since the active genes were different at different time points, these active PPI networks dynamically changed over time. The details of the dynamic network construction algorithm are shown in Algorithm 1.
**Algorithm 1** Dynamic Network Construction**Input:** A static PPI (S_PPI) network represented as Graph G = (V, E, W), a time series of the gene expression profile of each gene in G, parameter k.**Output:** The active networks of each time point.Step1: Identify two categories of genes, the time-dependent genes and the time-independent genes. using Equations (1) and (2), according to their expression profiles.Step2: Filter out the noise genes in the time-independent genes.Step3: Identify the active genes of each time point from the remaining two categories of genes by judging whether or not their expression values are above the threshold (calculated by Equation (3)).Step4: Map the active genes of each time point to the S_PPI network and extract the active networks of each time point.

#### 2.2.2. Fusing the Active Protein–Protein Interactive Networks of Each Time Point

After constructing the active networks of each time point, the next step was to fuse them into a single network, which captured the shared and complementary network structure of all the active networks, offering insight into how the expression of proteins was similar across different time points from the view of the network structure. To formally define the process of fusing networks, the following variables were introduced.

A static PPI network (S_PPI) can be represented as an undirected graph G = (V, E, W), where a node *v*∈V represents a gene and an edge *e*(*u*,*v*) ∈E denotes an interaction between two genes *v* and *u*. *w*(*u*,*v*) denotes the weight of the edge *e*(*u*,*v*), which measures the similarity between genes v and u. A dynamic PPI can be represented as a serial of active networks of different time points G_1_, G_2_,.... G_i_, … G_n_, where G_i_ = (V_i_, E_i_, W_i_) represents a subgraph of G at the *i*th time point. V_i_∈V is the set of nodes that are active at the *i*th time point. E_i_∈E is a subset of E that connects the active genes at the *i*th time point. W*_i_* is an adjacency matrix of G_i_, where its entry w*_i_*(*u_i_*,*v_i_*) measures the closeness of two nodes in the *i*th active network. The edges in the active network of each time point are weighted by Equation (5).
(5)$$w_{i}(u_{i},v_{i})=\exp(\frac{ECC_{i}(u_{i},v_{i})}{\mu *\varepsilon_{u_{i},v_{i}}})$$
(6)$$ECC_{i}(u_{i},v_{i})=\frac{\left|N_{i}(u_{i})\cap N_{i}(v_{i})\right|}{\min(N_{i}(u_{i})-1,N_{i}(v_{i})-1)}$$
(7)$$\varepsilon_{u_{i},v_{i}}=\frac{mean(ECC_{i}(u_{i},N_{i}(u_{i})))+mean(ECC_{i}(v_{i},N_{i}(v_{i})))+ECC_{i}(u_{i},v_{i})}{3}$$
where *N_i_*(*u_i_*) is the neighbor of *u_i_* in G_i_. *ECC_i_*(*u_i_*,*v_i_*) is the edge-clustering coefficient of *e_i_*(*u_i_*,*v_i_*) in G_i_, which is defined as the number of common neighbors of node *u_i_* and node *v_i_* in G_i_ divided by the number of common neighbors that might possibly exist between them. Since essential genes tend to form density clusters [22], their edge clustering coefficients can describe the degree to which two genes tend to cluster together. Similar to previous works [22,23,46,47], the edges in the active networks of each time are weighted by the edge clustering coefficients (see Equation (5)). Mean (*Ecc_i_*(*u_i_*,*N*(*u_i_*))) is the average of the edge clustering coefficient values between *u_i_* and its neighbors in G_i_. *μ* is a parameter that is empirically set to 0.5 according to the recommendation in Reference [35].

For the active network of the *i*th time point G_i_, its adjacency matrix *W_i_* has two derivatives, namely, matrix *P_i_* and matrix *S_i_*. Matrix *P_i_* carries the global information about the similarity of each gene to all the others obtained by performing normalization on *W_i_*:(8)$$p_{i}(u_{i},v_{i})=\{\begin{array}{cc} \frac{w_{i}(u_{i},v_{i})}{2\sum_{k_{i}\neq u_{i}}w_{i}(u_{i},k_{i})} & ,v_{i}\neq u_{i}\\ 1/2 & ,v_{i}=u_{i} \end{array}$$

Matrix *S_i_* only encodes the similarity between each gene in G_i_ and its *K* nearest neighbors (*K* = 20 according to the recommendation in Reference [35]):(9)$$s_{i}(u_{i},v_{i})=\{\begin{array}{cc} \frac{w_{i}(u_{i},v_{i})}{2\sum_{k_{i}\in N_{i}(u_{i})}w_{i}(u_{i},k_{i})} & ,v_{i}\in N_{i}(u_{i})\\ 0 & ,\mathrm{otherwise} \end{array}$$

Given the number M of active networks at different time points, we could construct an adjacency matrix *W_i_* of G_i_ using Equation (5) for the *i*th time point, *i* = 1, 2, 3, 4, … *M*. *P_i_* and *S_i_* were obtained from Equations (8) and (9), respectively. The aim of the network fusion was to fuse the *M* active networks into a single network. The process was as follows.

Firstly, the similarities between any two networks were calculated based on the Euclidean distance of their adjacency matrixes *W_i_* (*i* = 1, 2, 3 … *M*). Then the nearest two networks, i.e., *i* and *j*, were selected to fuse by the following iterative process.
(10)$$P_{i}^{t+1}=S_{i}*P_{j}^{t}*{\left(S_{i}\right)}^{T}$$
(11)$$P_{j}^{t+1}=S_{j}*P_{i}^{t}*{\left(S_{j}\right)}^{T}$$

Let $P_{i}^{0}=P_{i}$ and $P_{j}^{0}=P_{j}$ represent the initial two statuses at iteration step *t* = 0. $P_{i}^{t+1}$ and $P_{j}^{t+1}$ represent the status matrix of the active networks at the *i*th and the *j*th time point after t iteration steps, respectively. After *t* iteration steps, the fused network of the two networks was computed as
(12)$$R=\frac{P_{i}^{t}+P_{j}^{t}}{2}$$

Then, R was the result of the fused active networks *i* and *j.* After that, the similarities between R and the remaining active networks were recomputed again. R and its closest active network were selected to fuse into one network by repeating the above process until all the active networks were fused into a single network. Algorithm 2 shows the algorithm for fusing active PPI networks.
**Algorithm 2** Active PPI network fusion**Input:** Active networks of each time point, parameter K.**Output:** Final fused network.Step1: Construct adjacency matrix *W_i_* of the *i*th active network (I = 1, 2, 3, 4, …, M) using Equation (5).Step2: Construct *P_i_* and *S_i_* of the *i*th active network using Equations (8) and (9).Step3: Calculate the similarities between any two networks based on the Euclidean distance of their adjacency matrixes.Step4: Select the nearest two active networks *G_i_* and *G_j_*, Pi0=Pi, Pj0=Pj, *t* = 0.Step5: Compute Pit+1 and Pjt+1 using Equations (10) and (11), let *t* = *t* + 1.Step6: Repeat step 5 until *t* = 20.Step7: Compute the fused network R of *G_i_* and *G_j_* using Equation (12).Step8: Let *W_r_* = R, construct *P_r_* and *S_r_* of the fused network R using Equation (8) and (9).Step9: Find the nearest active network *G_k_* to R from the remaining active networks, let Pk0 = Pk, *t* = 0.Step10: Compute Prt+1 and Pkt+1 using Equations (10) and (11), let *t* = *t* + 1.Step11: Repeat step 10 until *t* = 20.Step12: Compute the fused network of R and *G_k_* using Equation (12), the fused network is named as R.Step13: Remove *G_k_* from active network list and repeat steps 8 to 12 until all the active networks are fused to a final network.Step14: Output the final fused network.

#### 2.2.3. Ranking Genes in the Fused Network

After fusing the active networks of different time points, an algorithm was designed to assign each gene in the fused network a ranking score. The ranking score measured the importance of the gene in the fused network from both the global and local perspectives.

A random walking process was implemented on the fused network to capture the global information of each gene. Let H be an F*F adjacency matrix of the final fused network. All its entries, i.e., *h*(*i*,*j*), were normalized by row. F is the number of genes in the network. In fact, F is the number of genes that are active at one of the time points. Let *pr*(*i*) be the ranking score of node *i* with respect to its global property in the fused network, which can be computed as follows.
(13)$$pr(i)=(1-a)*d(i)+a\sum_{j\in N(i)}h(i,j)pr(j)$$
(14)$$d(i)=\frac{o(i)}{\max_{i\in F}(o(i))}$$
where *o*(*i*) denotes the orthologous scores of node *i*, which is calculated by the number of times that the node has orthologs in the reference organisms. $\max_{i\in F}(o(i))$ is the maximal orthologous score among all the nodes in the network. Similar to Reference [22], we adopted an iterative process to numerically solve Equation (13). Here, parameter а was set to 0.5 according to the recommendation in Reference [22].

The interaction frequency entropy (IFE) of a gene in the final fused network measured its local topological properties. For a gene *i*, since we only considered its local properties, the interactions connecting to its *K* closest neighbors were selected to calculate its *IFE* values (*K* = 20 according to the recommendation in Reference [35]).
(15)$$IFE(i)=-\sum_{k\in KNN(i,K)}\frac{h(i,k)}{\left|KNN(i,K)\right|}\log(\frac{h(i,k)}{\left|KNN(i,K)\right|}),\;k\neq i$$
(16)$$IFE(i)=\frac{IFE(i)-\min(IFE)}{\max(IFE)-\min(IFE)}$$
where *KNN*(*i*,*K*) denotes the *K* closest neighbor set of node *i*, |*KNN*(*i*,*K*)| denotes the selected neighbor set size. Equation (16) was employed to perform the min-max normalization on the node’s *IFE* value.

Eventually, the ranking score of a node *i* in the final fused network, which was represented by FDP(*i*), equaled to the linear combination of its global topological score denoted as *pr*(*i*) and its local closest neighbors’ influence denoted as *IFE*(*i*). The parameter *λ* (0 ≤ *λ* ≤ 1) was used to adjust the weight of the two scores in the ranking score. Algorithm 3 shows the algorithm for computing the FDP values of genes.

FDP(*i*) = *λ***pr*(*i*) + (1 − *λ*) *IFE*(*i*)
(17)
**Algorithm 3** FDP**Input:** A static PPI network represented as Graph G = (V, E, W), gene expression profile, orthologs data sets between Yeast and 99 other organisms (ranging from *H**.sapiens* to *E**.coli*), stopping error ε, parameter а, λ.**Output:** FDP values of genes.Step1: Construct active networks of each time point using the Dynamic Network Construction algorithm.Step2: Fuse these active networks into a final fused network using the Active PPI network fusion algorithm.Step3: Calculate the orthologous scores of each node in the final fused network using Equation (14).Step4: Construct matrix *H* and normalize all its entries by row.Step5: Initialize *pr* with *pr*^0^ = *d*, let *t* = 0.Step6: Compute *pr^t+^*^1^ using Equation (13), let *t* = *t* + 1.Step7: Repeat step 6 until ||prt−prt−1||1≤ε.Step8: Calculate the IFE value of each gene in the final fused network using Equations (15) and (16).Step9: Calculate the FDP value of each gene in the final fused network by linearly combining its pr value and IFE value (see Equation (17)).

## 3. Results

In order to evaluate the performance of FDP in essential gene prediction, we compared the FDP with other existing methods (DC [10], BC [11], CC [12], SC [13], EC [14], IC [15], NC [16], PeC [18], ION [22], APPIN_DC [32], and APPIN_NC [32]). DC, BC, CC, SC, EC, IC, and NC are typical centrality-based methods that only consider the topological properties of genes in the S_PPI network. PeC and ION are two methods based on the S_PPI network that combine gene expression profiles or orthologous information with the S_PPI network. APPIN_DC and APPIN_NC are two methods based on the D_PPI network constructed using Xiao’s method [32]. The parameters in ION were selected according to the authors’ suggestion. All genes in the PPI network were ranked in descending order according to their ranking scores computed by the FDP, as well as other methods that were compared. After that, the top 100, 200, 300, and 400 of the ranked genes were selected as the candidates for essential genes. The performance of each method was judged according to how well the predicted genes matched the known genes. This evaluation method has been widely used in previous research procedures [16,18,22,48].

In this section, we first discuss the effect of parameter λ on the performance of the FDP. Then we compared the FDP with the other existing methods. After that, the results of the FDP and the other existing methods were analyzed in detail.

### 3.1. Effects of Parameter λ

In the FDP, parameter λ regulates the contribution of global network diffusion and local interaction frequency entropy when predicting essential genes based on the fused dynamic networks. This section focuses on the prediction accuracy analysis for parameter λ with different values, ranging from 0 to 1. When λ was set to 0, the ranking scores were calculated considering only the local topological properties of genes. When λ was set to 1, the ranking scores were calculated considering only the global topological properties of genes. The detailed results based on the DIP_PPI network and SC_net networks are listed in Table 2 and Table 3, respectively. Here, the parameter T was the number of selected candidate essential genes, ranging from 100 to 400. The prediction accuracy was measured in terms of the number of true essential genes in candidates.

Table 2 and Table 3 showed that the performance of the FDP based only on the local topological properties (λ = 0) is very poor. It was because the final fused network was a fully connected graph, which would introduce many false positive connections. However, the performance of the FDP considering the global topological properties rose sharply, because the orthologous properties of genes in the global topological property scores made a great contribution to ranking real essential genes. The performance of the FDP where it only combined the genes’ orthologous property with the genes’ global topological property (λ = 1) achieved the best performance when predicting a small number of essential genes. However, it was slightly poorer than the performance that considered both the local and global topological properties (λ ranging from 0.8 to 0.9), with an increase in the number of candidate genes selected. The reason may have been that the essential genes with high orthologous scores tended to rank in the top place, whilst the essential genes with high local centrality scores tended to rank at a slightly lower place. Consequently, we set λ to 0.8 in this work to make the FDP achieve good performance when predicting both a small and large number of essential genes.

### 3.2. Comparing with Other Methods

To assess the prediction performance of the FDP, the number of real essential genes identified by the FDP and other existing methods were compared, when the various top numbers of ranked genes were selected as candidates. Figure 2 and Figure 3 illustrate the results based on the DIP_PPI network and the SC_net network, respectively.

By selecting the top 100 of genes, the FDP achieved an 89% and 90% prediction accuracy on the DIP_PPI and SC_net networks, respectively. This was a 14% and 20% improvement compared to the ION, which had the best performance amongst all the other methods being compared on the two corresponding networks. When the top 200 genes were selected, the prediction accuracy of the FDP achieved about 82% accuracy on the two networks, which was nearly 10% higher than the ION. When the top 300 of genes were selected as candidates, the FDP still had a nearly 75% prediction accuracy on the two networks, which was 3% and 2% higher than the ION on the DIP_PPI and SC_net networks, respectively. When selecting the top 400 of genes, the FDP had a comparable prediction performance to the ION.

PeC predicts essential genes by integrating gene expression profiles with the static PPI network. Compared with PeC, when selecting the top 100, top 200, top 300, and top 400 of proteins as candidates, the accuracies of the FDP improved by 20.3%, 18%, 11.5%, and 11.1%, respectively, on the DIP_PPI network, and the accuracies improved by 23.3%, 20.9%, 22.8%, and 22.4%, respectively on the SC_net network. As for APPIN_NC, which predicts essential genes based on a dynamic PPI network and edge-clustering coefficient, in each top number of selected genes, the performance of the FDP was 30.9%, 26.2%, 27.4%, and 30.2% higher than that of the APPIN_NC on the DIP_PPI network, and it was 25%, 21.8%, 27.7%, and 22.8% higher than that of the APPIN_NC on the SC_net network. NC had the best performance among the seven centrality methods based on the static PPI network (DC, BC, CC, SC, EC, IC, and NC). Compared to the NC, in each top number (top 100, top 200, top 300, and top 400), the prediction accuracy of the FDP improved by 61.82%, 30.16%, 22.53%, and 21.74% on the DIP_PPI network, respectively, and it improved by 16.88%, 13.29%, 13%, and 12% on the SC_net network, respectively. Hence, overall, the FDP outperformed all the other comparative methods in the prediction of essential genes. Especially, with the small number of candidate genes selected, the advantage of the FDP becomes increasingly obvious.

### 3.3. Evaluation in Terms of Jackknife Curves

To investigate the performance of all the testing methods when selecting the different number of genes ranked at the top as candidates, jackknife curves were employed to show the results, where the *x*-axis represents the number of genes ranked at the top in descending order, according to their ranking scores computed by the corresponding methods. The *y*-axis is the cumulative count of the real essential genes within the ranked genes. Figure 4a,b illustrate the jackknife curves of all the methods based on the DIP_PPI network and the SC_net network, respectively. The two figures show that the FDP dramatically outperformed the methods based on the centrality of the static PPI network, such as the DC, IC, EC, SC, BC, NC, and CC. The FDP also outperformed the methods based on the centrality of the dynamic PPI network, such as the APPIN_DC and APPIN_NC. The FDP consistently exceeds the PeC which identifies essential genes by integrating gene expression data with the static PPI data. Compared with the ION that identifies the essential genes by integrating orthologous information with the static PPI data, the FDP also achieved better prediction performance when selecting less than 400 candidate genes. With more candidates selected, the curves of the two methods were very close.

### 3.4. Evaluation in Terms of Precision-Recall Curve

Precision-recall (PR) curves were also plotted to further show the overall performance of the comparative methods. Precision measures the percentage with which the predicted essential genes match the known genes in all the predicted genes. Recall measures the percentage that known essential genes matched the predicted ones over all the known essential genes. Figure 5a,b illustrate the PR curves of all the methods based on the DIP_PPI network and SC_net network, respectively. The figures show that the PR curves of the FDP are clearly above the curves of all the other methods on both the DIP_PPI network and SC_net network.

## 4. Conclusions

Essential genes play important roles in cell life under certain conditions and their mRNA expression levels tend to change within a narrow range. Under these observations, in this work, a novel method was proposed to identify essential genes by fusing the dynamic PPI networks of different time points. Compared with previous methods, our method hierarchically fuses the active networks of different time points into a single one. Moreover, it comprehensively utilizes the genes’ orthologous property and both their global and local topological properties to select the candidate essential genes from the fused network. The prediction results on two yeast PPI network datasets, show that our method improves essential gene prediction significantly, compared to the methods based on the static PPI network, including the methods considering the topological properties, i.e., DC, NC, and also the methods combining the PPI network with other biological properties, i.e., PeC and ION. Moreover, our method also outperformed the methods based on Xiao’s dynamic PPI network [4], i.e., APPIN_DC and APPIN_NC. All the results indicated that fusing the dynamic PPI networks and combining proteins’ orthologous properties with the PPI network improved the performance in the prediction of essential genes.

Compared with the existing methods, the FDP shows outstanding performance when selecting a small number of genes as the candidate essential genes. It may benefit from the construction of a dynamic network, which filters out the non-active genes of each time point. However, some real essential genes that consistently express low values across different time points have also been regarded as noise and have been ignored. It causes the decrease of the FDP’s prediction performance when selecting a large number of candidates. Hence, our future work is to construct a high quality dynamic network from the expression profiles that are full of mRNA isoforms and inevitable background noise. The prediction of essential genes also has great relations with the biological properties of known essential genes. New potential correlations between biological events and essential genes will be mined, such as alternative splicing. Moreover, the fused network is fully connected, which introduces some false interactions between the genes and causes poor performance when only considering the topological properties in the network. Therefore, another future work for us is to develop a more efficient strategy to fuse the active networks of different time points.

## Figures and Tables

**Figure 1 genes-10-00031-f001:**
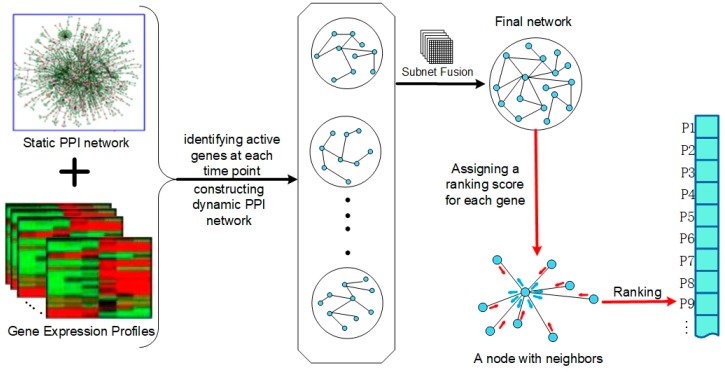
The workflow of our method.

**Figure 2 genes-10-00031-f002:**
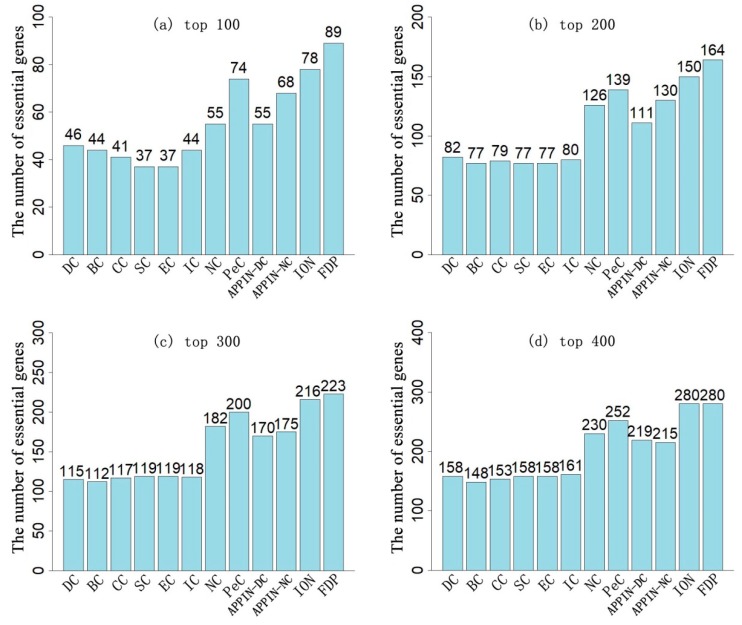
Comparison of the number of essential genes identified by the FDP and other existing methods based on the DIP_PPI network. (**a**–**d**) respectively show the results of these methods when selecting the top 100, 200, 300, and 400 of the ranked genes as candidate essential proteins. The data labels above the bars are the number of true essential proteins identified by the corresponding methods in each top number of ranked genes.

**Figure 3 genes-10-00031-f003:**
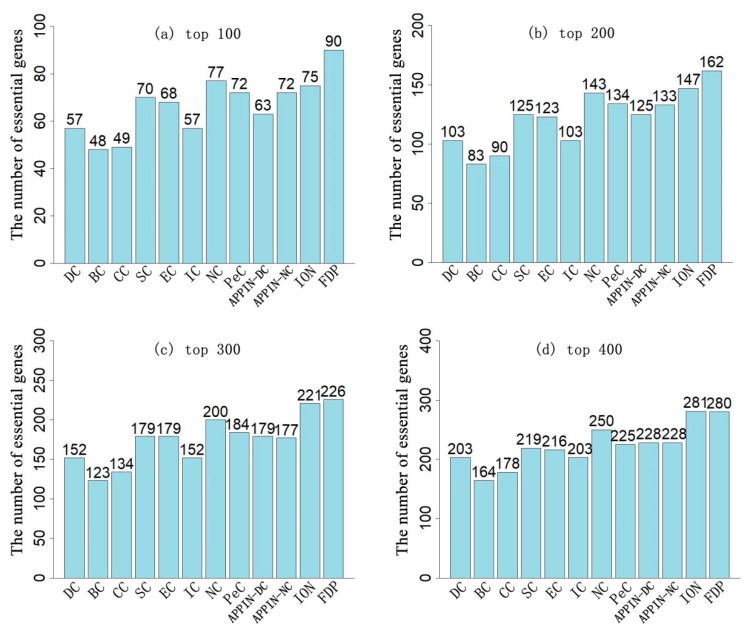
Comparison of the number of essential genes identified by the FDP and other existing methods based on the SC_net network. (**a**–**d**) respectively show the results of these methods when selecting top 100, 200, 300, and 400 of the ranked genes as candidate essential proteins. The data labels above the bars are the number of true essential proteins identified by corresponding methods in each top number of ranked genes.

**Figure 4 genes-10-00031-f004:**
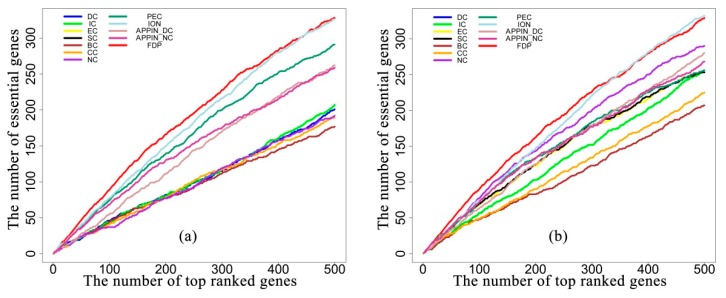
Jackknife curves of the FDP and other existing methods based on the DIP_PPI network (**a**) and the SC_net network (**b**).

**Figure 5 genes-10-00031-f005:**
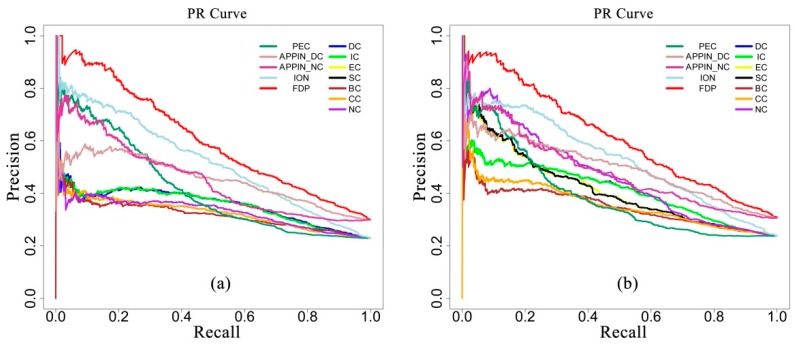
Precision-recall (PR) curves of the FDP and other existing methods based on the DIP_PPI network (**a**) and SC_net network (**b**).

**Table 1 genes-10-00031-t001:** Details of the two different yeast data sets. PPI: protein–protein interactive

Network	Genes in S_PPI	Edges in S_PPI	Genes in D_PPI	Essential Genes in S_PPI	Essential Genes in D_PPI
DIP_PPI	5093	24743	2759	1167	827
SC_net	4746	15166	2559	1130	785

**Table 2 genes-10-00031-t002:** Effects of parameter λ on the performance of the FDP based on the DIP_PPI network.

T	0	0.1	0.2	0.3	0.4	0.5	0.6	0.7	0.8	0.9	1
100	47	70	79	82	85	87	90	90	89	90	92
200	108	111	130	147	151	154	159	163	164	165	168
300	156	166	176	193	201	211	215	215	223	229	226
400	201	212	225	230	242	252	249	273	280	285	277

**Table 3 genes-10-00031-t003:** Effects of parameter λ on the performance of the FDP based on the SC_net network.

T	0	0.1	0.2	0.3	0.4	0.5	0.6	0.7	0.8	0.9	1
100	37	58	77	82	88	88	90	91	90	91	90
200	87	100	134	147	152	154	158	163	162	164	167
300	131	156	182	199	209	217	222	222	226	232	221
400	176	199	224	248	255	266	269	275	280	277	272

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
