# Peer review of "A Novel Method for Identifying Essential Genes by Fusing Dynamic Protein–Protein Interactive Networks"

_genes, 2019, doi:10.3390/genes10010031_

Round 1
Reviewer 1 Report
The authors propose in this work a novel method to identify the essential genes through the dynamic fusion of PPI networks. Based on the results shown by the authors, the proposed system improves its performance against other techniques when the number of candidate essential genes is small.
The paper lacks computational depth. No details are provided on how the algorithms have been implemented, what programming language has been used or what development environment has been selected.
It is necessary to include more information about the data set used in the experimentation process. What happens when a larger set of candidate essential genes is selected?
In figures 2 and 3 it is necessary to include in the legend a description for the images a, b, c and d.
In addition, there are many oral English language in the paper, such as "we” in lines 18,22,34,104, etc.
Author Response
We would like to thank you for your helpful comments and constructive suggestions and corrections. We have carefully read them and revised our manuscript based on these comments. In the attached document, we give our point-by-point responses to the reviewers' comments. We copied the original comments from the reviewers, and made them italic while responses are given in roman. We wish that our revised manuscript and responses fully address the points raised by the reviewers.

Reviewer 2 Report
I do not recommend the manuscript for publication in its current form. The presentation is very hard to follow due to very poor writing. The authors should significantly improve their writing and should ask for assistance from a language specialist. Many sentences are incomprehensible, and look almost like idiomatic expressions, translated word-for-word to English. Besides, the text is full of typos, missing spaces, etc. The authors should print the manuscript and read it carefully before submission.
The science look presentable, however, it is difficult to follow because the manuscript is saturated with equations and methods which can be moved to the Supplementary material. In a number of occasions an equation is referenced before being introduced. That should be corrected also.
Author Response

(The authors gave the same response as above.)

Round 2
Reviewer 2 Report
The writing has not been improved. Many sentences in the Introduction and Conclusion are barely understandable. That should be addressed because I believe the data is presentable to wider audience.
I have a few comments I would like to have addressed:
1. The first sentence in the Materials and Methods is misleading somehow. It says “The experiments were carried out on S. cerevisiae (Baker’s yeast).” The sentence implies that the authors carried our real world experiments, while their manuscript shows that they performed complex data analysis. That should be addressed. Maybe the authors should use other word than experiments, because experiment is usually perceived as something done in real world.
2. The authors use Gaussian noise to interpret noise in their input data. Although being a very reasonable approach, other types of noise may emerge in living systems, for instance if gene expressions have nonlinear relationships. Furthermore, if gene expression profiles are time dependent, the noise(s) should have variable properties as functions of time.
3. The authors use the Euclidian distance to measure adjacency matrix similarities. Can they justify the usage of that norm because other measures are often used in data analysis. For instance, norms based on absolute values are more sensitive to small fluctuations, while Euclidian distance (RMSD) is more sensitive to outliers. It would be good to give an overview of general gene expression behavior. On the other hand, the Euclidian norm performs well with the Gaussian noise. Maybe that is the reason.
4. Authors use relatively simple model system. In higher organisms gene expression is often guided by alternative splicing. That may affect the definition of essential genes. It would be good if the authors could be able to address that issue, and the applicability of their method to gene expression profiles in higher organisms in general. I do not expect authors to conduct additional data analysis, but they can devote a couple of sentences to those issues, for instance in Conclusion.
5. On the second page the authors use the abbreviation ION. I could not find its full form.
Author Response
Answers to Reviewer 2
The writing has not been improved. Many sentences in the Introduction and Conclusion are barely understandable. That should be addressed because I believe the data is presentable to wider audience.
Author’s Response: Thanks for your point. We have revised the sentence in the Introduction and Conclusion to make them easy to understand. We have read the manuscript carefully again and revised the typos and syntax errors. Moreover, the manuscript has been double checked by the person who is proficient in English. All revisions in the manuscript have been highlighted in the submission.
1. The first sentence in the Materials and Methods is misleading somehow. It says “The experiments were carried out on S. cerevisiae (Baker’s yeast).” The sentence implies that the authors carried our real world experiments, while their manuscript shows that they performed complex data analysis. That should be addressed. Maybe the authors should use other word than experiments, because experiment is usually perceived as something done in real world.
Author’s Response: Thanks for your point. In our manuscript, the experiment refers to perform FDP and other computational methods on a dataset to test their prediction performance. To avoid misleading, we replaced the word “experiment” with “prediction” or removed it. We also revised the sentence “The experiments were carried out on S. cerevisiae (Baker’s yeast).” into following sentence.
“FDP as well as the other existing computational methods are applied to predict essential genes of S. cerevisiae (Bakers’ Yeast).”
2. The authors use Gaussian noise to interpret noise in their input data. Although being a very reasonable approach, other types of noise may emerge in living systems, for instance if gene expressions have nonlinear relationships. Furthermore, if gene expression profiles are time dependent, the noise(s) should have variable properties as functions of time.
Author’s Response: Thanks for your point. If gene expression profiles are time dependent, the noise(s) should have variable properties as functions of time. Sometimes, our method can filter out this kind of noises.
In this work, the genes are divided into two categories: time-dependent genes and time-independent genes. A gene is supposed to be time dependent if its gene expressions have linear relationships and can be modeled by an AR model of order p (see Equation(1)). A gene is regarded to be time independent if its gene expressions have nonlinear relationships (see Equation (2)). The expression profile of a gene will be considered as noise if the gene is time independent but also the mean of its expression values across all time points is very small (less than 0.5 according to the analysis in [4] ). To construct dynamic PPI network, the active genes of each time point should be extracted. The active genes of each time point are identified by using a k-sigma method, where k-sigma method computes an active threshold for each gene according to the mean and standard deviation of its expression values. Therefore, the nodes in the active PPI network of each time point may be time dependent or time independent, but all of them should have relative high expression values. The genes whose expression profiles are time dependent will also be filtered out if their expression values below the thresholds that are set by considering the characteristics of their expression curves. However, it is hard for our method to filter out the noises that consistently express high values across different time points. Inevitable background noise exists in the gene expression profiles. It is still a challenge to effectively determine which genes are expressed or active based on noisy expression data when constructing dynamic PPI network.
In the revised manuscript, we add this in the conclusion section, which have been highlighted in the submission.
3. The authors use the Euclidian distance to measure adjacency matrix similarities. Can they justify the usage of that norm because other measures are often used in data analysis. For instance, norms based on absolute values are more sensitive to small fluctuations, while Euclidian distance (RMSD) is more sensitive to outliers. It would be good to give an overview of general gene expression behavior. On the other hand, the Euclidian norm performs well with the Gaussian noise. Maybe that is the reason.
Author’s Response: Thanks for your point. In this work, we mainly focus on fusing the active networks of each time point in a hierarchical way. Euclidian distance is used to measure the similarity of the two active networks. There are some other methods, i.e. Cosine Similarity, which can measure the similarity of two matrixes. We didn’t do further analysis to find out which one is the best. We use Euclidian distance because it is popular and it performs well.
According to your suggestion, we compared the prediction performance when using Euclidian distance or Cosine Similarity to measure the similarity of two different networks. Following tables show the prediction results of the two methods on DIP PPI network and SC_NET PPI network when selecting different λ values. As we can see from the tables, on the two PPI networks, the numbers of predicted real essential genes are very close when using the two different methods to measure the similarity of two active networks.
Methods,T,0,0.1,0.2,0.3,0.4,0.5,0.6,0.7,0.8,0.9,1
Euclidian distance,100,47,70,79,82,85,87,90,90,89,90,92
,200,108,111,130,147,151,154,159,163,164,165,168
,300,156,166,176,193,201,211,215,215,223,229,226
,400,201,212,225,230,242,252,249,273,280,285,277
Cosine Similarity,100,44,63,72,78,82,85,89,90,90,90,91
,200,85,106,133,152,156,160,162,162,165,168,169
,300,126,149,184,209,216,219,226,227,228,236,228
,400,171,199,227,239,252,262,265,268,285,290,283
Table 1 Effects of parameter λ and similarity methods on the performance of FDP based on DIP PPI network.
Methods,T,0,0.1,0.2,0.3,0.4,0.5,0.6,0.7,0.8,0.9,1
Euclidian distance,100,37,58,77,82,88,88,90,91,90,91,90
,200,87,100,134,147,152,154,158,163,162,164,167
,300,131,156,182,199,209,217,222,222,226,232,221
,400,176,199,224,248,255,266,269,275,280,277,272
Cosin Similarity,100,59,72,81,83,84,88,89,88,91,91,92
,200,115,125,143,158,163,167,167,167,167,168,168
,300,164,180,201,216,219,225,228,228,231,231,232
,400,225,238,252,258,260,267,275,278,287,286,281
Table 2 Effects of parameter λ and similarity methods on the performance of FDP based on SC_NET PPI network.
4. Authors use relatively simple model system. In higher organisms gene expression is often guided by alternative splicing. That may affect the definition of essential genes. It would be good if the authors could be able to address that issue, and the applicability of their method to gene expression profiles in higher organisms in general. I do not expect authors to conduct additional data analysis, but they can devote a couple of sentences to those issues, for instance in Conclusion.
Author’s Response: Thanks for your suggestion. In this work, our model is only applied on predicting essential genes of yeast. In the future, we will try to apply it on higher organisms, i.e. human. However, the main challenge comes from correctly determining which genes transcript into the mRNA isoforms in the mRNA expression profiles. In high eukaryotic cells, a gene includes several introns and exons and can transcript into different mRNA isoforms guided by alternative splicing. Therefore, a gene may code for different proteins under certain conditions, but also different genes can probably generate an mRNA isoform. In the conclusion section of our revised manuscript, we mentioned that our future work is to construct high quality dynamic network from the expression profiles that are full of mRNA isoforms and inevitable background noisy. Besides, there are probably some potential correlation between alternative splicing and essential genes. Some researchers have found that the selection of wrong splice sites causes human disease. Hence, another future work for us to improve the prediction of essential genes is to mind new potential correlation between biological events and essential genes, such as alternative splicing.
5. On the second page the authors use the abbreviation ION. I could not find its full form.
Author’s Response: Thanks for your point. For clearness, we rewrite the sentence on the second page as follows.
The original sentence:
Peng et al. [22] propose ION that integrates the orthology with PPI networks to predict essential genes.
The revised sentence:
Peng et al. [22] propose a method for predicting essential genes by integrating the orthology with PPI networks, named by ION.
